# Primary thermometry triad at 6 mK in mesoscopic circuits

Z. Iftikhar[1], A. Anthore[1], S. Jezouin[1], F.D. Parmentier[1], Y. Jin[1], A. Cavanna[1], A. Ouerghi[1], U. Gennser[1] & F. Pierre[1]

Quantum physics emerge and develop as temperature is reduced. Although mesoscopic electrical circuits constitute an outstanding platform to explore quantum behaviour, the challenge in cooling the electrons impedes their potential. The strong coupling of such micrometre-scale devices with the measurement lines, combined with the weak coupling to the substrate, makes them extremely difficult to thermalize below 10 mK and imposes *in situ* thermometers. Here we demonstrate electronic quantum transport at 6 mK in micrometre-scale mesoscopic circuits. The thermometry methods are established by the comparison of three *in situ* primary thermometers, each involving a different underlying physics. The employed combination of quantum shot noise, quantum back action of a resistive circuit and conductance oscillations of a single-electron transistor covers a remarkably broad spectrum of mesoscopic phenomena. The experiment, performed in vacuum using a standard cryogen-free dilution refrigerator, paves the way towards the sub-millikelvin range with additional thermalization and refrigeration techniques.

[1] Centre de Nanosciences et de Nanotechnologies, CNRS, Univ Paris Sud-Université Paris-Saclay, Université Paris Diderot-Sorbonne Paris Cité, 91120 Palaiseau, France. Correspondence and requests for materials should be addressed to F.P. (email: frederic.pierre@u-psud.fr).

Advances towards lower temperatures are instrumental in the fundamental exploration of quantum phenomena. In the context of quantum electronics, typical examples are the exploration of the correlated fractional quantum Hall physics[1–5], of the quantum criticality, for example, with multichannel Kondo nanostructures[6–8], or of the quantum aspects of heat[9–12]. Although commercial dilution refrigerators readily achieve temperatures in the 5–10 mK range at the mixing chamber, the pertinent value is the temperature of the electrons within the cooled quantum circuits. Owing to microwave heating, insufficient thermal contacts and electrical noise transmitted through the measurement lines, this electronic temperature is usually well above the refrigerator base temperature. Consequently, only rare examples demonstrate electronic temperatures significantly below 10 mK in quantum circuits. Moreover, the concept of temperature pervades the laws of physics, and its accurate knowledge is generally imperative whenever comparing experimental measurements with theoretical predictions; however, establishing the validity of the thermometry is particularly challenging already below 50 mK. Because of the thermal decoupling between electrons and substrate, it requires a comparison of the electronic temperature determined *in situ*, in the same device, by different methods.

The lowest electronic temperatures in solid-state quantum circuits were obtained in large, millimetre-scale, devices that are thereby weakly sensitive to heating through the measurement lines. The lowest reported value of 3.7 mK, to our knowledge, was obtained in a large array of 600 metallic islands, each $\sim 100\,\mu m$ wide and interconnected by tunnel junctions[13]. Comparably low temperatures, of possibly $\sim 4$ mK, were inferred in two-dimensional (2D) electron gas (2DEG) chips in the quantum Hall regime by two different teams[2,14,15]. For the more broadly pertinent micrometre-scale mesoscopic circuits, the reported electronic temperatures are significantly higher. We note the remarkably low value of 9 mK determined with current fluctuation measurements across a quantum point contact (QPC) in a 2DEG[16]. Although single-electron devices are particularly challenging, because of their high charge sensitivity, comparably low electronic temperatures, down to $\sim 10$ mK, were recently demonstrated in 2DEG quantum dots[6,8,17].

Here we investigate three primary electronic thermometers, and demonstrate quantum electronic transport at 6 mK in micrometre-scale mesoscopic circuits. For this purpose, the experiment is performed on a highly tunable 2DEG nano-structure, that can be set by field effect to different circuit configurations. The complementary underlying physics of the thermometry methods give us access to different facets of the electronic temperature, and cover a broad spectrum of mesoscopic quantum phenomena. Whereas quantum shot noise thermometry measures the temperature of the electronic Fermi quasiparticles, through their energy distribution[18], the quantum back action of a resistive circuit also probes the temperature of the electromagnetic environment[19]. In contrast, the temperature inferred from the conductance oscillations of a single-electron transistor (SET) is very sensitive to charge fluctuations induced by non-thermal high-energy photons[20]. At the applied magnetic field $B = 1.4$ T, we find with the quantum shot noise measured across a voltage-biased QPC $T_N \simeq 6.0 \pm 0.1$ mK. From the conductance peaks across the device set to a SET configuration, we obtain $T_{CB} \simeq 6.3 \pm 0.3$ mK. From the dynamical Coulomb blockade conductance dip across two separate realizations of a QPC in series with a resistance, we find $T_{DCBL} \simeq 6 \pm 1$ mK and $T_{DCBR} \simeq 6.5 \pm 1$ mK. The observed agreement between the three primary thermometers establishes their validity on an extended temperature range.

## Results

**Cooled tunable mesoscopic circuit.** A colourized electron micrograph of the measured device is shown in Fig. 1a, with the corresponding circuit schematic displayed Fig. 1b. A high-mobility 2DEG is located 105 nm below the surface of a Ga(Al)As heterojunction. It is confined by etching within the darker grey areas delimited by bright lines, and can be tuned *in situ*, by field effect, with the bias voltages applied to metallic gates deposited at the surface and capacitively coupled to the 2DEG (colourized green, yellow and blue in Fig. 1a). The metallic split gates at the top-left ($QPC_L$) and bottom-right ($QPC_R$) of Fig. 1a (colourized green) are used to form QPCs in the 2DEG. Note that the split gates at the top-right of Fig. 1a (colourized yellow) is here set to fully deplete the 2DEG underneath, thereby closing the gate, and can be ignored. The buried 2DEG is galvanically connected, with a negligible interface resistance, to the central micrometre-sized metallic island (colourized red). For this purpose, the metallic island was diffused into the Ga(Al)As heterojunction by thermal annealing. The lateral continuous gates at the surface (colourized blue) implement the equivalent of short-circuit switches in parallel with the island (blue switches in Fig. 1b). The experiments are performed with a magnetic field

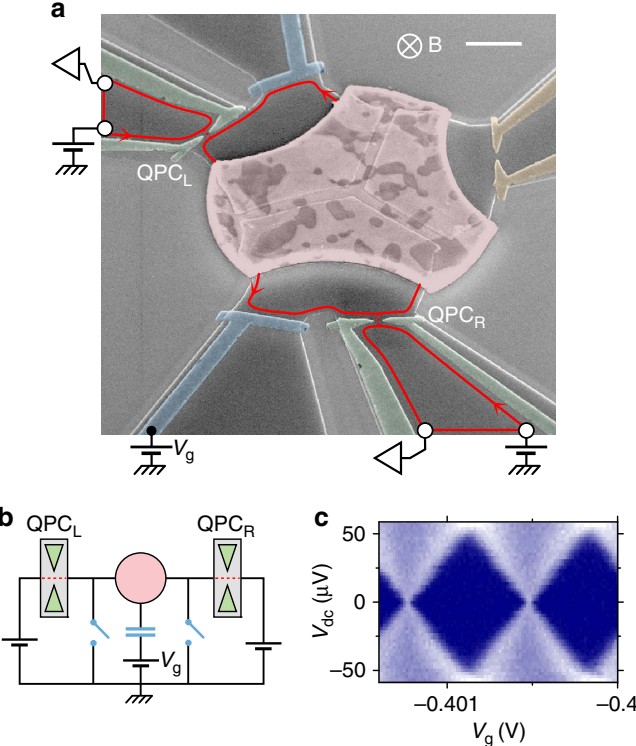

**Figure 1 | Cooled electrical nanostructure. (a)** Coloured micrograph of the measured device. Top-right scale bar length, 1 µm. The micrometre-scale metallic island (red) is connected to 200 µm wide electrodes (represented as white circles) through two QPCs (green split gates) formed in a buried 2DEG (darker grey). The lateral gates (blue) implement the switches shown in **b** by field effect. The sample is immersed in a magnetic field $B$ corresponding to the integer quantum Hall regime, with the current propagating along the edge (red lines) in the direction indicated by arrows. **(b)** Schematic electrical circuit. Using the switches, the same device can be tuned *in situ* into a voltage-biased QPC, a SET or a QPC embedded into a resistive circuit. **(c)** Charging energy characterization, for Coulomb blockade phenomena. With the device tuned into a SET, $E_C = 25 \pm 1\,\mu eV$ is obtained from the height of the diamond patterns in the SET conductance (larger values shown brighter) measured versus gate ($V_g$) and bias ($V_{dc}$) voltages.

$B$ applied perpendicular to the 2DEG, which corresponds to the quantum Hall regime at integer filling factors $v = 6$, 3 and 2 for $B = 1.4$, 2.7 and 3.8 T, respectively. In this regime the current flows along $v$ chiral edge channels, represented as a single red line with the propagation direction indicated by arrows in Fig. 1a. Note that the quantum Hall effect is not necessary for the investigated primary thermometers (although it allows eliminating possible heating artefacts in the quantum shot noise thermometry, see Discussion). An important device parameter is the single-electron charging energy $E_C \equiv e^2/2C$ of the central metallic island, with $C$ its overall geometrical capacitance and $e$ the elementary electron charge. In particular, $E_C$ sets the temperature scale extracted from Coulomb blockade thermometry. The charging energy is most straightforwardly determined by setting the device in the SET configuration, with the short-circuit switches open (as shown in Fig. 1a,b) and $QPC_{L,R}$ tuned to tunnel contacts. The SET conductance is plotted in Fig. 1c (higher values shown brighter) versus the capacitively coupled gate voltage $V_g$ and the applied drain-source dc voltage $V_{dc}$. The charging energy is directly related to the periodic 'Coulomb diamond' patterns in Fig. 1c: $E_C = |eV_{dc}^{max}|/2 \simeq 25 \pm 1 \mu\text{eV}$, with $V_{dc}^{max}$ the diamonds' maximum dc voltage.

**Electronic current fluctuations.** The current across a voltage-biased quantum coherent conductor fluctuates because of the thermal agitation (the Johnson-Nyquist noise) and the granularity of charge transfers (the shot noise)[18]. These fluctuations give information on the charge of the carriers, for example, in the fractional quantum Hall regimes[3,16,21,22], as well as on the statistics of the charge transfers[23–25], and also provide a very robust primary thermometer for the electronic temperature[26].

We have measured the current fluctuations across the device tuned into a voltage-biased QPC (schematic shown in top panel of Fig. 2a, see Supplementary Note 2 for details on the current fluctuations measurement set-up). For this purpose, the right short-circuit switch in Fig. 1b was effectively closed, by applying $V_g = 0$ to the continuous gate adjacent to $QPC_R$. Thereby, the 2DEG is not depleted and the edge current flows underneath the gate without back-scattering, implementing an ideal closed switch (see Supplementary Fig. 1).

The dependence with bias voltage $V_{dc}$ of the current fluctuations' spectral density, $S_I(V_{dc})$, is directly related to the electrons' energy distribution (including in out-of-equilibrium situations[27,28]). For a short quantum conductor, the excess spectral density $\Delta S_I(V_{dc}) \equiv S_I(V_{dc}) - S_I(0)$ can be calculated in the standard framework of the scattering approach[29–31]. It reads[18]:

$$\Delta S_I = \frac{2e^2}{h} \sum_n \tau_n(1 - \tau_n) \times \left[ eV_{dc} \coth\left(\frac{eV_{dc}}{2k_B T}\right) - 2k_B T \right], \quad (1)$$

where the quantum conductor is described as a set of independent conduction channels, indexed by the label $n$, each characterized by a transmission probability $\tau_n$, and with $k_B$ ($h$) the Boltzmann (Planck) constant. Note that the noise added by the amplification chain is cancelled out by considering the excess spectral density $\Delta S_I$. Importantly, the product between the gain of the amplification chain and $\sum \tau_n(1 - \tau_n)$ is given by the temperature-independent linear slope predicted at $|eV_{dc}| \gg k_B T$. Fitting the raw spectral density data based on equation 1 therefore allows a self-calibrated determination of the electronic temperature, without requiring the knowledge of $\{\tau_n\}$ or of the amplification gain (see Supplementary Note 2 for further details).

The symbols in the top panel of Fig. 2a display the excess current spectral density measured at $B = 1.4$ T versus the dc bias voltage applied across the QPC, which is tuned into the

advantageous configuration of a single half-transmitted conduction channel ($\tau \simeq 0.55$). Note that in order to display the current fluctuations data in A²/Hz, and, although it is not necessary for extracting the electronic temperature, the effective amplification chain gain is calibrated by matching the linear bias voltage increase in the raw spectral density at large $|eV_{dc}| \gg k_B T$ with the prediction of equation 1 for the measured $\tau = 0.55$. The continuous (dashed) line shows $\Delta S_I$ calculated using equation 1 with $\tau = 0.55$ and $T = 6.0$ mK ($T = 0$, with a negative vertical offset to match the $T = 6.0$ mK calculation at $|eV_{dc}| \gg k_B T$). Experimentally, the main difficulty is to reach a sufficient resolution to accurately extract the electronic temperature. To this aim, we developed a fully homemade cryogenic noise amplification scheme, based on high electron mobility transistors grown and nanostructured in the laboratory[32,33]. Despite the unfavourable current–voltage conversion at $v = 6$ because of the low quantum Hall resistance $h/6e^2 \simeq 4.3$ kΩ, we resolve $\Delta S_I$ with an extremely high statistical precision of $\pm 9 \times 10^{-32}$ A² Hz$^{-1}$, slightly smaller than the symbols' size.

Most directly, we have determined the electronic temperature and experimental uncertainty $T_N = 6.0 \pm 0.1$ mK from the mean value (red horizontal line in bottom panel of Fig. 2a) and statistical uncertainty of an ensemble of 131 values (symbols in bottom panel of Fig. 2a) independently obtained by separately fitting successive noise measurement sweeps $\Delta S_I(V_{dc})$. Note that the $\Delta S_I$ data shown in the top panel of Fig. 2a was obtained by averaging these successive sweeps (each resolved with an individual noise precision of $\pm 10^{-30}$ A² Hz$^{-1}$).

**Coulomb blockade oscillations.** At low temperatures, $T \ll E_C/k_B$, the charge of a mesoscopic island connected through tunnel contacts is quantized in units of the elementary electron charge $e$. This allows for the manipulation of single electrons in circuits, which has led to the field of 'single electronics'[19]. Setting the device in the SET configuration (see schematic in bottom panel of Fig. 2b), charge quantization results in periodic peaks of the SET conductance $G_{SET}$ when sweeping the capacitively coupled gate voltage $V_g$. In the presence of dc bias voltage, the peaks develop into periodic 'Coulomb diamond' patterns as shown in Fig. 1c. The width of these conductance peaks at zero dc bias voltage constitutes a well-known primary thermometer, frequently used in the context of mesoscopic physics. For a metallic island, with a continuous density of states and connected through tunnel contact, the SET conductance reads[34]:

$$G_{SET} = \frac{G_\infty}{2} \frac{2E_C(\delta V_g/\Delta)/k_B T}{\sinh(2E_C(\delta V_g/\Delta)/k_B T)}, \quad (2)$$

with $G_\infty$ the classical (high temperature) conductance of the SET, $\Delta \simeq 712 \pm 2 \mu$V the gate voltage period and $\delta V_g$ the gate voltage difference to charge degeneracy. Note that the Coulomb blockade thermometry is possible only with tunnel contacts. In the presence of connected conduction channels with large transmission probabilities, the quantum fluctuations of the island's charge would average out Coulomb oscillations and thereby impede the Coulomb blockade thermometry (see ref. 35 for a characterization of charge quantization versus transmission probability on the same device).

The symbols in the top panel of Fig. 2b represent $G_{SET}$ measured at $B = 1.4$ T versus $\delta V_g$. The continuous line shows the SET conductance calculated using equation 2 with $T = 6.3$ mK, $G_\infty = 0.088 e^2/h$, $\Delta = 711 \mu$V and $E_C = 25 \mu$eV.

Similarly to quantum shot noise thermometry, we determined the electronic temperature and statistical precision $T_{CB} = 6.3 \pm 0.05$ mK from an ensemble of 222 values (symbols in the bottom panel of Fig. 2b) obtained by separately fitting

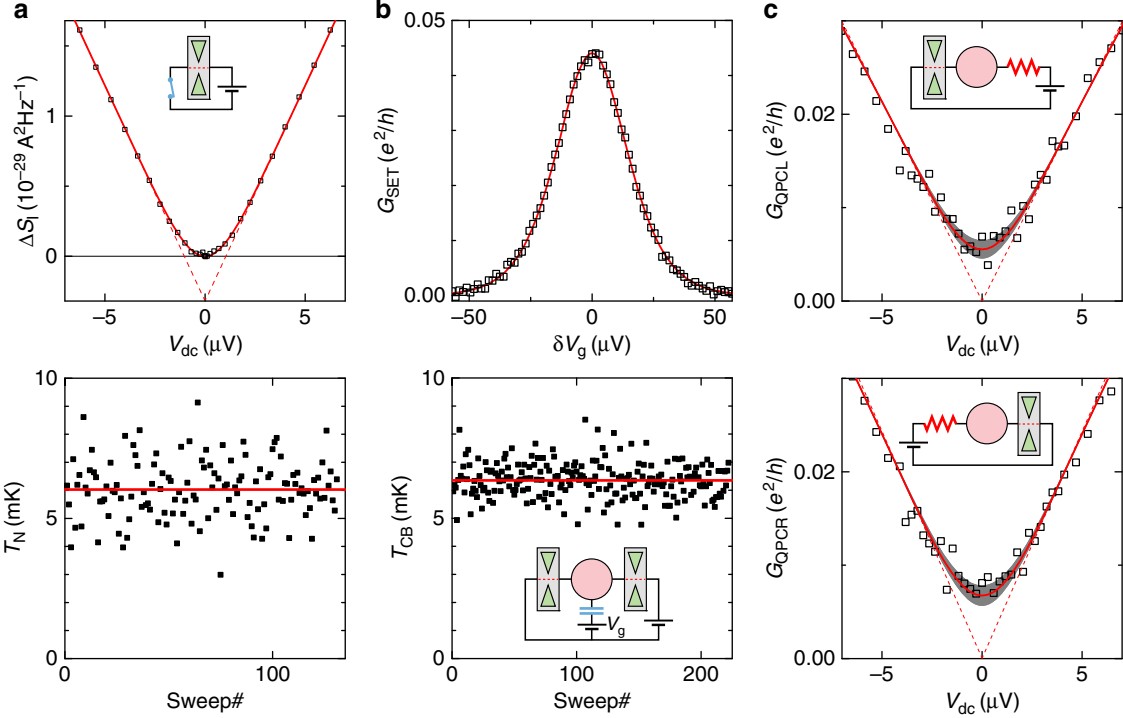

**Figure 2 | Primary electronic thermometry. (a)** Quantum shot noise. Symbols in the top panel represent the measured excess spectral density of the current fluctuations across a QPC biased with the dc voltage $V_{dc}$ (see configuration schematic). The red continuous (dashed) line is the calculated excess current fluctuations for $T_N = 6.0$ mK ($T_N = 0$, with a matching negative offset). In the bottom panel, the different electronic temperatures $T_N$ shown as symbols are each obtained by fitting a different (successive) voltage bias sweep of the quantum shot noise. From the statistical averaging of 131 values, we find $T_N \simeq 6.0 \pm 0.1$ mK (horizontal red line) with an accuracy comparable to the provisional low-temperature-scale standard (PLTS-2000). **(b)** Coulomb blockade. Symbols in the top panel represent the measured conductance $G_{SET}$ across the device tuned into a SET (see schematic in bottom panel) versus the gate voltage difference $\delta V_g$. The continuous line is the calculated conductance for $T_{CB} = 6.3$ mK. The different electronic temperatures $T_{CB}$ represented by symbols in the bottom panel are each obtained from a different gate voltage sweep $G_{SET}(\delta V_g)$. From the averaging of 222 values, we find $T_{CB} \simeq 6.3 \pm 0.05$ mK (horizontal red line). Note that the accuracy on $T_{CB}$ is limited to $\pm 0.3$ mK by our uncertainty on the charging energy, $E_C = 25 \pm 1$ μeV. **(c)** Dynamical Coulomb blockade. The electronic temperature $T_{DCB}$ is obtained by fitting the conductance $G_{QPCL,R}$ of $QPC_{L,R}$ (symbols) versus voltage bias with the dynamical Coulomb blockade theory in the presence of a known series resistance $R = h/2e^2$ (see configuration schematic). The dashed lines display the predicted suppression of the conductance at $T = 0$ and $eV_{dc} \ll E_C$, here linear in $V_{dc}$. We find $T_{DCB} \simeq 6 \pm 1$ mK ($6.5 \pm 1$ mK) for $QPC_L$ ($QPC_R$) from the fit shown as a continuous line in the top (bottom) panel. The estimated uncertainty of $\pm 1$ mK is displayed as a grey background.

individual sweeps of $G_{SET}(\delta V_g)$. The 222 sweeps are distributed among 14 adjacent Coulomb peaks, spreading over 10 mV in gate voltage. We find the same electronic temperature, at experimental accuracy, for the different Coulomb peaks and also for the 15 or 16 measurements of each peak. Note that our experimental accuracy $T_{CB} \simeq 6.3 \pm 0.3$ mK is limited by our resolution of the charging energy, $E_C \simeq 25 \pm 1$ μeV. The uncertainty is consequently much larger than the statistical precision. Note also that, despite a relatively low ac voltage of $0.35$ μ$V_{rms}$ applied to probe $G_{SET}$, we estimate (using the master equation generalizing equation 2 to finite voltages[19]) that it is responsible for an effective increase of 0.1 mK in $T_{CB}$ (we have not corrected for this small effect). Finally, we point out that the $G_{SET}$ data shown in the top panel of Fig. 2b were obtained by averaging the 222 individual sweeps.

**Dynamical Coulomb blockade conductance renormalization.** The conductance of a quantum coherent conductor is progressively reduced upon cooling by the quantum back action of the circuit in which it is embedded[19]. This phenomenon, called dynamical Coulomb blockade, results from the granularity of charge transfers combined with Coulomb interactions. It has been extensively studied, and the theory is now well established in the simplest limit of a small tunnel conductor inserted into a linear

circuit (see ref. 19 and references therein; for recent developments beyond the tunnel limit see refs 36–39).

We consider here the case of a tunnel contact in series with a linear resistance $R$, as shown in the schematics of Fig. 2c. In this configuration, the conductance at zero-bias voltage (zero temperature) vanishes with temperature $T$ (bias voltage $V_{dc}$) as $T^{2Re^2/h}$ (as $V_{dc}^{2Re^2/h}$). Similarly to quantum shot noise thermometry, the equilibrium ($V_{dc} \ll k_B T/e$) to non-equilibrium ($V_{dc} \gg k_B T/e$) crossover provides a primary electron thermometer. In general, the electronic temperature can be extracted by fitting the conductance versus dc voltage with the full quantitative numerical prediction of the dynamical Coulomb blockade theory (see ref. 40 for a formulation involving a single numerical integration). Note that the extracted electronic temperature reflects equally the thermal energy distributions of the Fermi electron quasiparticles, and of the bosonic electromagnetic modes of the quantum circuit. The dynamical Coulomb blockade was previously used to probe the non-Fermi energy distribution of electrons driven out-of-equilibrium in the presence of a thermalized $RC$ circuit[41].

In the low-temperature and low-bias voltage regime ($k_B T$, $e|V_{dc}| \ll E_C$), the primary dynamical Coulomb blockade thermometry reduces to the simple procedure described below. The QPC conductance at low temperature ($T \ll E_C/k_B$) and at

zero-bias voltage $V_{dc} = 0$ reads[42]:

$$G_{QPC}(T) = \frac{G_\infty \pi^{\frac{3Re^2}{h} + \frac{1}{2}} \Gamma\left(1 + \frac{Re^2}{h}\right)}{2\Gamma\left(1.5 + \frac{Re^2}{h}\right)} \left(\frac{Re^2}{h} \frac{k_B T}{E_C}\right)^{\frac{2Re^2}{h}}, \quad (3)$$

where $G_\infty$ is the tunnel conductance in the absence of dynamical Coulomb blockade renormalization and $\Gamma(x)$ is the gamma function. Extracting the temperature from the zero-bias conductance apparently requires a precise knowledge of both $G_\infty$ and the circuit parameters ($R$, $C$). However, the necessary information is provided by the bias voltage dependence. In the non-equilibrium regime $k_B T \ll e|V_{dc}|$ and at low energy compared with the single-electron charging energy $e|V_{dc}| \ll E_C$, the QPC conductance reads[19]:

$$G_{QPC}(V_{dc}) = \frac{G_\infty \left(\frac{\pi}{\gamma}\right)^{\frac{2Re^2}{h}} \left(\frac{2Re^2}{h} + 1\right)}{\Gamma\left(2 + \frac{2Re^2}{h}\right)} \left(\frac{Re^2}{h} \frac{e|V_{dc}|}{E_C}\right)^{\frac{2Re^2}{h}}, \quad (4)$$

with $\gamma \simeq \exp(0.5772)$. Consequently, the bias voltage exponent gives the series resistance $R$, and one can rewrite the zero-bias voltage conductance as:

$$G_{QPC}(T) = \frac{A(R)}{B(G_\infty, R, E_C)} (k_B T)^{\frac{2Re^2}{h}}, \quad (5)$$

with $B(G_\infty, R, E_C) \equiv G_{QPC}(V_{dc})/|eV_{dc}|^{2Re^2/h}$ calibrated from the conductance measured in the low-energy non-equilibrium regime, where equation 4 applies, and $A(R)$ a known function, straightforwardly obtained from equations 3 and 4.

Here we determined the electronic temperature by setting one QPC in the tunnel regime ($G_\infty \sim 0.1e^2/h$), while the other QPC was tuned to fully transmit two electronic channels, thereby implementing a linear series resistance $R = h/2e^2$ (which is not renormalized by dynamical Coulomb blockade[37–39,43]; obtained from a very broad and flat conductance plateau owing to the quantum Hall effect[11,39]). Symbols in the top (bottom) panels of Fig. 2c represent the conductance measured with the left (right) QPC in the tunnel regime, versus dc bias voltage, at $B = 1.4\,\text{T}$. The continuous lines display the quantitative numerical calculations of the dynamical Coulomb blockade prediction using the separately characterized $E_C = 25\,\mu\text{eV}$ and $R = h/2e^2$ (also corresponding to the linear bias voltage dependence), and with $G_\infty = 0.123e^2/h$, $T = 6\,\text{mK}$ for the top panel ($G_\infty = 0.139e^2/h$, $T = 6.5\,\text{mK}$ for the bottom panel). The grey areas represent a temperature uncertainty of $\pm 1\,\text{mK}$. The dashed lines are the $T = 0$ predictions of equation 4 for the same device parameters. Note that for the present circuit implementation $A(R = h/2e^2) \simeq 0.40$ and the non-equilibrium conductance increases linearly with bias voltage, as can be directly verified on the conductance data. See Supplementary Fig. 2 for a comparison between the numerically calculated dynamical Coulomb blockade predictions and the data up to larger bias voltages.

**Electronic temperature versus experimental conditions.** Information on the limiting factors towards lower electronic temperatures $T$ in our cryogen-free dilution refrigerator is obtained by measuring $T$ for different magnetic fields $B$ and for different additional Joule powers $P_J$ dissipated directly on the mixing chamber plate. Note that $T$ is here obtained from quantum shot noise thermometry up to 35 mK, and from the identical but faster readings of our standard $RuO_2$ thermometer at higher temperatures. Each set of symbols in Fig. 3 corresponds to a different applied $B \in \{1.41, 2.74, 3.76\}$ T. For $P_J \gtrsim 5\,\mu\text{W}$, we observe the usual quadratic dependence with temperature ($T^2 \propto P_J$), independently of the applied $B$. However, we find that the electronic temperature at zero Joule power is higher for larger

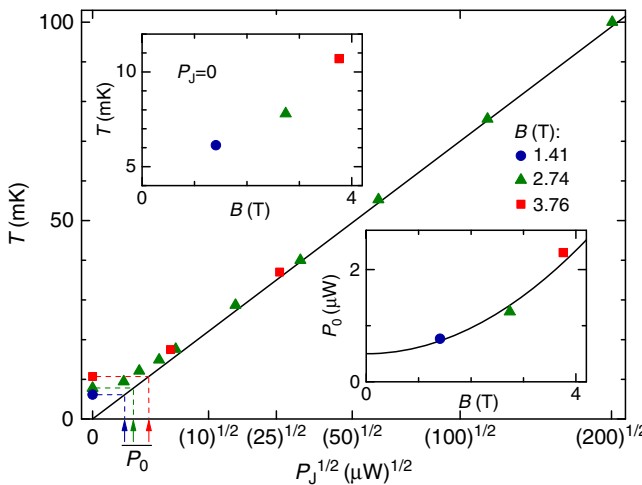

**Figure 3 | Temperature versus magnetic field and Joule heating.** The measured electronic temperature (symbols) is represented versus the square root of the Joule power $P_J$ dissipated on the mixing chamber of the dilution refrigerator, for three different values of the perpendicular magnetic field $B$. The continuous line displays $T = 7\sqrt{P_J/1\,\text{W}}$ K. Top left inset: the temperature at $P_J = 0$ (symbols) increases with the applied magnetic field $B$. Bottom right inset: the intrinsic dissipated power $P_0$, estimated assuming $T \propto \sqrt{P_0 + P_J}$ (arrows in main panel), is plotted as symbols versus magnetic field. The continuous line displays $P_0 = 0.5 + 0.1(B/1\,\text{T})^2$ $\mu$W.

magnetic fields (top left inset). Assuming that the observed relationship $T = 7\sqrt{P_J/1\,\text{W}}$ K (straight black line in the main panel) holds at all temperatures when substituting the additional Joule power by the full dissipated power $P = P_0 + P_J$, we extract the refrigerator-dissipated power $P_0$ versus magnetic field. The corresponding $P_0$ values are shown as symbols in the bottom right inset. We find that the increase of $P_0$ with $B$ is compatible with a quadratic magnetic field dependence (continuous black line: $P_0 = 0.5 + 0.1(B/1\,\text{T})^2$ $\mu$W), which is a typical signature of eddy current dissipation.

## Discussion

A 6 mK electronic temperature was obtained in micrometre-scale quantum circuits using a medium-sized cryogen-free dilution refrigerator (Oxford instruments Triton, with 200 μW of cooling power at 100 mK), with the sample in vacuum and in the presence of a 1.4 T magnetic field. At larger magnetic fields $B$, we observe a temperature increase that corresponds to an additional dissipated power quadratic in $B$, as typically expected for eddy currents. In our cryogen-free refrigerator, the underlying vibrations originate from the pulse tube.

The sample environment and wiring shown Fig. 4 offers a proven guideline to ultra-low electronic temperatures with an all-purpose set-up, including 35 measurement lines and a top-loaded sample holder. Although additional details are provided in the Supplementary Note 1, we here briefly point out several key ingredients. The sample is strongly protected from spurious high-energy photons, by two shields at base temperature. The most important thermal anchoring of the measurement lines at base temperature is performed by dipping insulated copper wires into silver epoxy very close to the sample, inside the inner stainless steel shield. The measurement lines are all individually shielded in a coaxial cable geometry (except for the above-mentioned copper wires and for a short distance inside the shielded sample holder, between the input connector and the RC filters). The high-frequency filtering and

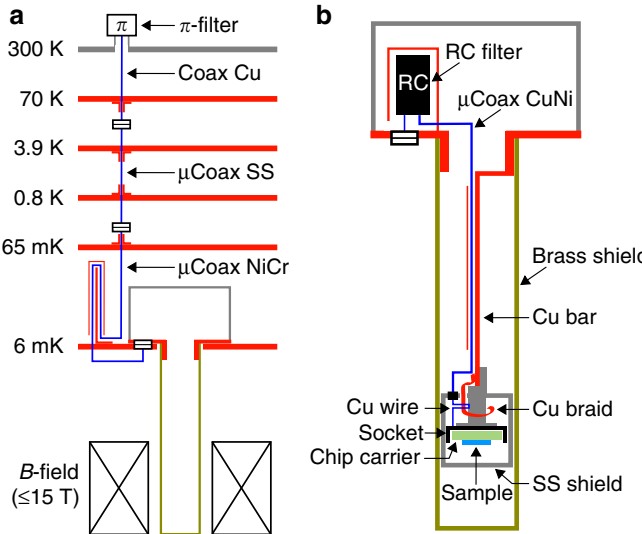

**Figure 4 | Experimental set-up.** (a) Diagrammatic representation of the electrical lines between room temperature (∼300 K) and the top-loaded sample holder at base temperature. (b) Schematic representation of the top-loaded sample holder.

initial thermalization to the mixing chamber plate of the electrical lines are performed with homemade resistive microcoaxes (μCoax NiCr in Fig. 4a)[20]. Because the electrical noise integrated over the full bandwidth needs to be smaller than a fraction of μV, we only keep the bandwidth used for the measurements with personalized RC filters directly located inside the sample holder. This is most particularly important with a cryogen-free dilution refrigerator in the presence of a magnetic field because of the electrical noise induced by vibrations. We now compare the three investigated primary electronic thermometers.

Quantum shot noise thermometry stands out as the most robust and straightforward approach. It is based on simple physics, directly probes the temperature of the electrons through their energy distribution[18] and does not require a separate calibration of the noise measurement set-up. The main possible artefact is local heating induced by the dissipated Joule power at finite dc bias. Such a heating typically scales linearly with $V_{dc}$ (refs 44,45). It is therefore difficult to distinguish from a slight increase in the shot noise[24,25]. The present implementation in the quantum Hall regime, however, provides a strong protection against heating artefacts, owing to the spatial separation between incoming and outgoing currents. Although it is not necessary to determine the factor $\sum \tau_n(1 - \tau_n)$ for the voltage-biased quantum conductor, it is important to make sure that it does not depend on $V_{dc}$. For a single-channel quantum conductor, the dependence of $\tau(1 - \tau)$ with voltage bias is minimized at $\tau \sim 0.5$, and $\tau(V_{dc})$ can be monitored simultaneously with the noise measurements. The main challenge with quantum shot noise thermometry is in the sensitivity of the noise measurement set-up; however, the associated temperature uncertainty can be statistically quantified. Note that the achieved resolution of $6.0 \pm 0.1$ mK is comparable to the accuracy of the provisional low-temperature-scale standard (PLTS-2000)[46].

Coulomb blockade thermometry is also very straightforward and has the advantage of being less demanding on the measurement sensitivity. It is consequently widespread in the field of mesoscopic physics. However, the extracted temperature is easily/often artificially increased by charge fluctuations in the device vicinity, or by the electrical noise on the capacitively coupled gates. Such an artefact could be detected

as a gate voltage-dependent increase in the noise level, proportional to $\partial G_{SET}/\partial V_g$, provided that a significant part of the charge fluctuations is within the measurement bandwidth. In general, the corresponding temperature increase is difficult to establish, except by comparing with another electronic thermometer. Here the agreement obtained with both the quantum shot noise and dynamical Coulomb blockade thermometers demonstrates a negligible artificial increase in the electronic temperature.

Dynamical Coulomb blockade thermometry can be difficult to use in general, if the surrounding circuit is not known *a priori* at relevant GHz frequencies. As in the case of quantum shot noise, a possible artefact is heating at a finite dc bias. This can be minimized by using a tunnel contact of very large impedance compared with the circuit. In contrast to quantum shot noise, the quantum Hall regime does not provide a protection against heating (in the central metallic island, for the dynamical Coulomb blockade experimental configurations). However, the very large renormalized tunnel resistance, 100 larger than the series resistance, ascertains negligible heating effects. Moreover, the dynamical Coulomb blockade thermometry is here particularly straightforward to implement because of the precise knowledge of the circuit.

With the consistent temperatures obtained by three primary thermometers, each relying on different physical mechanisms, we firmly established electronic thermometry standards in the regime of ultra-low temperatures. The achievement of 6 mK electronic temperature, with the mesoscopic circuit in vacuum and using a medium-sized dilution refrigerator, provides a platform for further reduction of the temperature, using additional thermalization and cooling techniques[15,47,48] towards the sub-millikelvin range.

## Methods

**Sample.** The sample was nanostructured by standard e-beam lithography in a Ga(Al)As 2DEG of density $2.5 \times 10^{11}$ cm$^{-2}$ and mobility $10^6$ cm$^2$ V$^{-1}$ s$^{-1}$. The AuGeNi metallic island was diffused by thermal annealing into the semiconductor heterojunction to make an electrical contact of negligible resistance with the 2DEG (see Methods in ref. 8 for the electrical characterization of the contact in the same sample).

**Measurement techniques.** The differential conductance measurements were performed using standard lock-in techniques at frequencies below 200 Hz and using rms ac excitation voltages smaller than $k_B T/e$. The sample was current-biased by a voltage source in series with a 100 MΩ polarization resistance at room temperature. The applied current was converted on-chip into a voltage independent of the device configuration by taking advantage of the well-defined quantum Hall resistance to an adjacent grounded electrode ($h/\nu e^2$ at filling factor $\nu$). Similarly, the current transmitted across (reflected from) the device was converted into a voltage with the $h/\nu e^2$ quantum Hall resistance. The noise measurement set-up includes a homemade cryogenic pre-amplifier and an $L$–$C$ tank circuit of resonant frequency 0.84 MHz; see the online Supplementary Information and also the Supplementary Material of ref. 11 for a more detailed description.

**Data availability.** The data that support the findings of this study are available from the corresponding author upon request.

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

## Acknowledgements

This work was supported by the European Research Council (ERC-2010-StG-20091028, no. 259033), the French RENATECH network, the national French programme 'Investissements d'Avenir' (Labex NanoSaclay, ANR-10-LABX-0035) and the European Seventh Framework Program (EU FP7, no. 263455).

## Author contributions

Measurements and analysis: F.P. with inputs from Z.I.; low temperature set-up: F.P. with inputs from A.A. and S.J.; noise measurement set-up: F.D.P., S.J., A.A. and F.P.; HEMT nanofabrication: Y.J.; heterojunction growth: A.C., A.O. and U.G.; sample nanofabrication: F.D.P. and A.A; manuscript preparation: F.P. with inputs from A.A. and U.G.; project planning and supervision: F.P.

## Additional information

**Competing financial interests:** The authors declare no competing financial interests.

