## [Peer review file · Nature Communications]

Reviewers' Comments:

Reviewer #1 (Remarks to the Author)

The authors present high quality manuscript on a low temperature setup specially adapted for cryogenics to millikelvin temperatures of nanoscale circuits. I expect that setups, such as the one discussed in this paper, will open up possibilities for new physics in the near future, thus I think this manuscript provides a new impetus to such efforts.

This work is an experimental tour-de-force in low temperature techniques. The two notable achievements are:

1. The cooling of electricity confined to nanoscale devices to 0.006 K. This is an important achievement on its own as cooling nanoscale electrical circuits is fraught with additional difficulties as compared to cooling larger samples.
2. Authors have a novel setup which allows temperature measurement using three different techniques, based on different physical phenomena. This is achieved by employing an innovative on-chip capability of reconfiguring the electric pathways to different configurations using electrical gates.

The paper is well organized and well written; it contains a detailed description of the setup and measurement techniques, a careful analysis of the data. While the temperature measuring techniques used are not new, the simultaneous use of the three temperature measurements techniques combined with cooling of the system to 0.006 K is a significant development in the field of quantum electronics.

Nonetheless, there are a small number of changes I ask the authors to implement before I can recommend publication:

1. While an expert can figure out where etching was done on the sample shown in Fig.1a, the contrast between the etched and unetched regions is poor. A better coloring scheme is necessary.
2. When it comes to noise, in a few places in the manuscript the authors use a terminology which is misleading. In the bottom of page 1 "... 9mK determined with quantum shot noise measurements..." and on page 2, under second subtitle under Results: "Electronic quantum shot noise". In fact shot noise carries no information on temperature, Johnson noise does. These two types of noises are intertwined in mesoscopic conductors. Nonetheless, temperature sensitivity is solely due to the Johnson noise. For clarity, please replace "shot noise" in the above phrases with either "current fluctuations" or "Johnson-Nyquist noise".
3. Clarification needed: the resistance of the QPC for the noise data shown in Fig.2a, as calculated at $V=0$ using the Johnson noise expression, appears to be 104 kohm. How is this related to the statement that the QPC is set to a "single half-transmitted channel"?
4. I request additional data, perhaps included in the Supplement. For noise measurements, please include spectral information. Such data will be valuable in assessing vibration issues in cryofree systems.
5. Pertaining to Fig.1c, please include a statement or data on what is the range of gate voltages the Coulomb diamonds develop at and if the temperatures obtained at different gate voltages studied are the same.

Finally, I have a request. If the authors find it reasonable, I ask them to include some additional references. The authors talk about difficulties in reaching temperatures below 10 mK and mention two groups succeeding doing this in fractional quantum Hall systems. Those setups have already yielded new physics, can the authors consider citing one or two. In addition, in APL 102, 243102 (2013) the on-chip multiplexer presented shares common features with the technique used in this manuscript.

Reviewer #2 (Remarks to the Author)

The authors use a simple mesoscopic circuit at low temperature to test three (to some extent) different electronic probes for the device temperature. The goal is to achieve an as low as possible electron temperature using appropriate filters and thermal anchoring. Here, the authors show that they reach 6 mK with all three thermometers yielding consistent results. The thermometers are based on measuring the shot noise, the width of a narrow Coulomb blockade (CB) peak and the shape of the zero-bias anomaly for "weak tunneling" where the conductance dips due to so-called dynamical CB.

While it is a nice tour de force demonstration that all thermometers show the same, I do not see real novelty here. All three systems (thermometer principles) have been used and have been described very well. The first two methods, noise and CB resonance, are widely used for thermometry. The last one is a bit newer, but I do not see what it adds in effect to the rest. Similarly, while 6 mK is a low electron temperature, it is not a record. One also has to see that there is a large middle contact that can act as an efficient heat sink, allowing to reach this temperature without too much problems.

In conclusions, it is a nice demonstration of thermometry at low temperatures, but there are no real new elements, no record in the lowest achieved electron temperature, and there is also no new physical insight that the experiments provide. I therefore cannot recommend this paper for Nature Communications which explicitly states that a successful paper should provide "... an advance in understanding likely to influence thinking in the field".

Reviewer #3 (Remarks to the Author)

The cooling of mesoscopic circuits, based on two dimensional electron gases and exploiting electrostatic quantum confinement, into the low mK temperature regime is a topic of significant current interest. It allows the exploration of electronic correlation effects, of quantum heat flow, and the prospect of new quantum electronic devices.

The difficulty of cooling the electron gas, due to strong thermal decoupling from the lattice, means that the electronic temperature will not necessarily reflect the temperature of the low temperature platform to which the device is mounted.

This provides a strong motivation to develop ways to directly measure electron temperature. This is the subject of this manuscript. The nanostructure they study is of ingenious design, and can be biased into three distinct configurations, to permit three independent measures of temperature: quantum shot noise; Coulomb blockade and dynamical Coulomb blockade.

The work reported is a variant on an earlier paper (Science) in which quantum heat flow was studied. Of course that work also required direct measurement of electronic temperature. On balance the present manuscript adds significantly to their previous published work, and is probably of sufficient interest and novelty to justify publication in Nature Communications.

For this referee, understanding the technical details of the present manuscript relied too heavily on reading the Science paper and its Supplementary Information, and study of the literature. I therefore recommend adding material, in some cases to the manuscript and perhaps also to the Supplementary Information, in order that the manuscript is adequately self-contained, and to better establish validity.

The study of quantum shot noise in semiconductor mesoscopic structures in the quantum hall regime has a distinguished historical record. I do believe that the discussion of electronic quantum shot noise given on p2/3 is incomplete. In equation (1) the voltage independent offset term (referred to but not written down) corresponds to the equilibrium Johnson/Nyquist noise from the QPC; it is temperature independent, [reference 17, equation 62]. Therefore, surely, it should be discussed, and its inclusion in the fitting procedure to determine temperature described.

In general there are also a few points that the authors take somewhat for granted, that will not be fully appreciated by the non-specialist reader, and I urge that these be rectified also (see below). (If the following comments contain some misunderstandings, let that in part only reinforce this

point).

I recommend publication, subject to these remarks, and the following detailed comments, being taken into account.

More detailed comments

It would be helpful to have a clear statement for the general reader, explaining why it is the case that it is essential to bias the 2DEG on a quantum Hall plateau. As I understand it, the quantum Hall voltage is measured and thus converts the noise current to a measured voltage noise. How the noise precision of 10^{-30} A²/Hz is obtained should be explicitly discussed.

The measurement set-up in the Science paper shows a tuned resonant across the input of the voltage amplifier. Is such circuit present in this case? What is the central frequency and bandwidth of the noise measurements reported?

The clear discussion of accuracy and precision for CBO is not replicated in the discussion of quantum shot noise. Factors limiting accuracy are not properly discussed e.g. what is the accuracy with which the transmission coefficient of the QPC is determined? It would also be helpful to explicitly state acquisition times needed to achieve the reported precision.

The section on dynamical coulomb blockade thermometry is more technical, and hard to follow in this account.

The details on how the sample is thermalized to the mixing chamber should be simply and explicitly stated; it is difficult to extract this crucial information from the technical details provided.

I question the value of including the section of p5 on "electronic temperature vs. experimental condition". Fig 3. plots the electronic temperature vs power to the mixing chamber. To compare the temperature measured in the mesoscopic device with that measured by a reliable independent thermometer (not a secondary resistance thermometer) on the mixing chamber over a wide temperature range would be valuable, but this was not available. [Elsewhere the paper concentrates on a cross-calibration at a single temperature]. Here, merely to extract the heat leak to the mixing chamber as a function of magnetic field is just not of sufficient interest for the main text of a Nature Communication. [The caption to Fig 3 refers to "electronic" temperature. Measured by which technique?]

Finally, to reinforce an earlier point, no detail is given on the noise thermometry set-up in this manuscript. I recommend adding this detail, in main text and in supplementary information. The criterion to provide sufficient information that the measurement could be reproduced, particularly important if the claim is to demonstrate new thermometry techniques, is not satisfied by this manuscript.

It is also worth stating that the reliance on being in the quantum Hall regime probably makes this technique less generic and more niche than the impression the reader would get from reading the abstract. Perhaps the abstract should reflect this.

To repeat, I recommend publication subject to the above issues being satisfactorily addressed.

We would like to thank the reviewers for their in-depth and constructive remarks.

Our answers to the specific points raised by the reviewers are inserted within their comments below.

Reviewer #1:

We thank the reviewer for his positive appreciation of our work and for his pertinent suggestions.

"The authors present high quality manuscript on a low temperature setup specially adapted for cryogenics to millikelvin temperatures of nanoscale circuits. I expect that setups, such as the one discussed in this paper, will open up possibilities for new physics in the near future, thus I think this manuscript provides a new impetus to such efforts.

This work is an experimental tour-de-force in low temperature techniques. The two notable achievements are:

1. The cooling of electricity confined to nanoscale devices to 0.006 K. This is an important achievement on its own as cooling nanoscale electrical circuits is fraught with additional difficulties as compared to cooling larger samples.

2. Authors have a novel setup which allows temperature measurement using three different techniques, based on different physical phenomena. This is achieved by employing an innovative on-chip capability of reconfiguring the electric pathways to different configurations using electrical gates.

The paper is well organized and well written; it contains a detailed description of the setup and measurement techniques, a careful analysis of the data. While the temperature measuring techniques used are not new, the simultaneous use of the three temperature measurements techniques combined with cooling of the system to 0.006 K is a significant development in the field of quantum electronics.

Nonetheless, there are a small number of changes I ask the authors to implement before I can recommend publication:

1. While an expert can figure out where etching was done on the sample shown in Fig.1a, the contrast between the etched and unetched regions is poor. A better coloring scheme is necessary."

The contrast was improved in Fig. 1a and Supplementary Fig. S1a.

"2. When it comes to noise, in a few places in the manuscript the authors use a terminology which is misleading. In the bottom of page 1 "... 9mK determined with quantum shot noise measurements..."

and on page 2, under second subtitle under Results: "Electronic quantum shot noise". In fact shot noise carries no information on temperature, Johnson noise does. These two types of noises are intertwined in mesoscopic conductors. Nonetheless, temperature sensitivity is solely due to the Johnson noise. For clarity, please replace "shot noise" in the above phrases with either "current fluctuations" or "Johnson-Nyquist noise".

We have followed the reviewer's recommendation to use the more generic formulation "current fluctuations", including also in the caption of Fig. 2a. The temperature information is indeed not in the pure shot noise at large bias voltage (nor in the pure Johnson-Nyquist noise at zero bias, as it is completely subtracted from the excess noise considered here, see also answer to point 3) but in the crossover from low to large bias.

"3. Clarification needed: the resistance of the QPC for the noise data shown in Fig.2a, as calculated at $V=0$ using the Johnson noise expression, appears to be 104 kohm. How is this related to the statement that the QPC is set to a "single half-transmitted channel"?"

The reviewer's remark points to a previously confusing offset in the current fluctuations (in Fig. 2a as well as in Eq. 1), see also the related answer to reviewer #3. We have now corrected this by using the excess current fluctuations ΔS_I , with respect to $S_I(V=0)$, both in Eq. 1 and for the measurements shown in the top panel of Fig. 2a. The previously subtracted voltage independent offset included a portion of the full Johnson-Nyquist noise, which quantitatively explains the apparent discrepancy observed by the reviewer.

"4. I request additional data, perhaps included in the Supplement. For noise measurements, please include spectral information. Such data will be valuable in assessing vibration issues in cryofree systems."

We have added the new Supplementary Note 2 and Supplementary Fig. S3, which describe much more precisely the current fluctuations measurement setup (including the used frequency window, as specifically requested by the reviewer). See also the related answer to reviewer #3.

"5. Pertaining to Fig.1c, please include a statement or data on what is the range of gate voltages the Coulomb diamonds develop at and if the temperatures obtained at different gate voltages studied are the same."

We now explicitly point out below Eq. 2 that the Coulomb blockade thermometry is only possible with sufficiently low transmission (tunnel, $\tau < \sim 0.1$) contacts, and we have added the reference 35 to a submitted work where the influence of transmission probability on Coulomb blockade oscillations is specifically studied on the same sample. Regarding the second point, we indeed do find the same electronic temperature for 14 different Coulomb peaks spreading over an explored gate voltage range V_g of 10 mV (each of the individual values of T_{CB} shown in the bottom panel of Fig. 2b correspond to one of the 14 Coulomb peaks; the 14 peaks were each measured 15 or 16 times to get the total number of 222 T_{CB} values). This is now explicitly mentioned in the last paragraph of the section "Coulomb blockade oscillations".

"Finally, I have a request. If the authors find it reasonable, I ask them to include some additional references. The authors talk about difficulties in reaching temperatures below 10 mK and mention two groups succeeding doing this in fractional quantum Hall systems. Those setups have already yielded new physics, can the authors consider citing one or two. In addition, in APL 102, 243102 (2013) the on-chip multiplexer presented shares common features with the technique used in this manuscript."

Following the reviewer's suggestion, we have added the new reference Ref. 5 to the previously cited Ref. 2.

Reviewer #2 :

We thank the reviewer for pointing that this experiment "is a nice tour de force" and that it provides a "demonstration of thermometry at low temperature".

"The authors use a simple mesoscopic circuit at low temperature to test three (to some extent) different electronic probes for the device temperature. The goal is to achieve an as low as possible electron temperature using appropriate filters and thermal anchoring. Here, the authors show that they reach 6 mK with all three thermometers yielding consistent results. The thermometers are based on measuring the shot noise, the width of a narrow Coulomb blockade (CB) peak and the shape of the zero-bias anomaly for "weak tunneling" where the conductance dips due to so-called dynamical CB.

While it is a nice tour de force demonstration that all thermometers show the same, I do not see real novelty here. All three systems (thermometer principles) have been used and have been described very well. The first two methods, noise and CB resonance, are widely used for thermometry. The last one is a bit newer, but I do not see what it adds in effect to the rest."

We agree with the reviewer that at least two of the three thermometers are widely used. In our view, as for metrology, this makes it even more important to experimentally establish their validity

and accuracy in the ultra-low temperature regime addressed by the present work. In practice, as pointed out in the article (end of first paragraph), this is very challenging because the thermal decoupling between electrons and substrate requires comparing the electronic temperature determined in-situ, in the same device, by different methods.

“Similarly, while 6 mK is a low electron temperature, it is not a record.”

We also agree that 6 mK is not the lowest electronic temperature ever reported in solid-state circuits; lower values, down to 3.7 mK, were achieved in millimeter-scale devices [13]. However, to our knowledge, the presently demonstrated 6 mK is a record for the very pertinent micrometer-scale mesoscopic circuits. Note that for the present shot noise thermometry the probed quantum conductor is a very small (~100 nm long) voltage biased quantum point contact. Even in the Coulomb blockade or dynamical Coulomb blockade configurations, the quantum circuits probed here are only a few micrometers, more than two orders of magnitude smaller than the millimeter-scale devices where a lower electronic temperature was previously achieved. In order to remove a possible confusion regarding the size of Ref. 13’s device (between ~100 μm and several millimeters), we have modified the corresponding description in the article (see page 1, right column, first sentence).

“One also has to see that there is a large middle contact that can act as an efficient heat sink, allowing to reach this temperature without too much problems.”

Note that the central micrometer-scale metallic island in our device cannot play any role (including heat sinking) in the quantum shot noise thermometry, as it is located downhill the QPC (the chiral edge channels arriving at the QPC are directly emitted from the voltage biased electrodes, see Supplementary Figure S1a).

In principle some cooling (heat sink) could indeed take place through the electron-phonon coupling in the central island for the Coulomb blockade and dynamical Coulomb blockade thermometry. Unfortunately, it is not expected to be significant at such low temperatures with our micrometer-scale metallic island (which has a volume 7 orders of magnitude smaller than the 600 interconnected metallic islands in the device of Ref. 13). Indeed, from the electron-phonon coupling determined in Ref. 11 for a similar device, and assuming that the phonons are at $T=0$, we obtain the negligible upper bound of $4 \cdot 10^{-20}$ W for the heat sinking at 6 mK electronic temperature.

We decided not to add a specific discussion as we do not make any claim regarding a possible heat sinking effect of the central island. If, in view of our answer, the reviewer believes we should mention explicitly that heat sinking at the central island does not apply (shot noise thermometry) or is expected to be negligible, we will make the modification.

“In conclusions, it is a nice demonstration of thermometry at low temperatures, but there are no real new elements, no record in the lowest achieved electron temperature, and there is also no new physical insight that the experiments provide. I therefore cannot recommend this paper for Nature Communications which explicitly states that a successful paper should provide "... an advance in understanding likely to influence thinking in the field".”

Reviewer #3 :

We thank the reviewer for pointing out the current general interest in “cooling mesoscopic circuits” and in the associated thermometry methods, and for his pertinent comments which have helped us to improve the manuscript.

“The cooling of mesoscopic circuits, based on two dimensional electron gases and exploiting electrostatic quantum confinement, into the low mK temperature regime is a topic of significant current interest. It allows the exploration of electronic correlation effects, of quantum heat flow, and the prospect of new quantum electronic devices.

The difficulty of cooling the electron gas, due to strong thermal decoupling from the lattice, means that the electronic temperature will not necessarily reflect the temperature of the low temperature platform to which the device is mounted.

This provides a strong motivation to develop ways to directly measure electron temperature. This is the subject of this manuscript. The nanostructure they study is of ingenious design, and can be biased into three distinct configurations, to permit three independent measures of temperature: quantum shot noise; Coulomb blockade and dynamical Coulomb blockade.

The work reported is a variant on an earlier paper (Science) in which quantum heat flow was studied. Of course that work also required direct measurement of electronic temperature. On balance the present manuscript adds significantly to their previous publish work, and is probably of sufficient interest and novelty to justify publication in Nature Communications.

For this referee, understanding the technical details of the present manuscript relied too heavily on reading the Science paper and its Supplementary Information, and study of the literature. I therefore recommend adding material, in some cases to the manuscript and perhaps also to the Supplementary Information, in order that the manuscript is adequately self-contained, and to better establish validity.”

In response to the reviewer, we now provide much more technical details, in particular in the new Supplementary Fig. S3 and Supplementary Note 2.

“The study of quantum shot noise in semiconductor mesoscopic structures in the quantum hall regime has a distinguished historical record. I do believe that the discussion of electronic quantum

shot noise given on p2/3 is incomplete. In equation (1) the voltage independent offset term (referred to but not written down) corresponds to the equilibrium Johnson/Nyquist noise from the QPC; it is temperature independent, [reference 17, equation 62]. Therefore, surely, it should be discussed, and its inclusion in the fitting procedure to determine temperature described.”

The reviewer points out the lack of discussion regarding a voltage independent offset not included in Eq. 1. Note that this is directly related to the remark 3 of reviewer #1. We agree with the reviewer that our presentation was not optimal, and switched to use the excess noise with respect to $V_{dc}=0$ both in the top panel of Fig. 2a and in Eq. 1. Importantly, as now obvious with the use of the excess noise, the overall offset in the spectral density of current fluctuations (vastly dominated by the noise added by the amplification chain) does not play any role in the extraction of the electronic temperature. The detailed fitting procedure is now very explicitly described in Supplementary Note 2. Note that in our previous Science paper Ref. 11, where the base electronic temperature was 22 mK, we used a markedly different thermal noise thermometry strategy, which is not primary as it required a very careful and demanding calibration of the amplification chain based on the temperature readings of a standard RuO₂ thermometers at “high” temperatures $T > 50$ mK. As we now very explicitly point out in the article (see page 4, left column) and detail in Supplementary Note 2, here the procedure is much simpler and robust since we don’t need to calibrate the amplification chain and we don’t rely on its perfect stability over long periods of time.

“In general there are also a few points that the authors take somewhat for granted, that will not be fully appreciated by the non-specialist reader, and I urge that these be rectified also (see below). (If the following comments contain some misunderstandings, let that in part only reinforce this point).

I recommend publication, subject to these remarks, and the following detailed comments, being taken into account.

More detailed comments

It would be helpful to have a clear statement for the general reader, explaining why it is the case that it is essential to bias the 2DEG on a quantum Hall plateau. As I understand it, the quantum Hall voltage is measured and thus converts the noise current to a measured voltage noise.”

It is not essential to be on a quantum Hall plateau, as now explicitly pointed out (section “Cooled tunable mesoscopic circuit”, page 2, right column), but it has the advantage to completely eliminate local heating artifacts that could arise in the shot noise thermometry (see the “Discussion” section). However other strategies are possible to minimize these artifacts. We find that the quantum Hall resistance is a convenient way to convert the current fluctuations into voltage fluctuations, but it is absolutely not necessary. In other labs this conversion is most often done by adding a resistor in series between the sample and the electrical ground.

“How the noise precision of 10^{-30} A²/Hz is obtained should be explicitly discussed”

We now provide many details regarding the noise measurement setup, which allows us to achieve such a high noise resolution. This is done both in the article and in the Supplementary Note 2, including a description of how the gain is calibrated (which is necessary to display the data in A²/Hz but not to extract the electronic temperature).

“The measurement set-up in the Science paper shows a tuned resonant across the input of the voltage amplifier. Is such circuit present in this case? What is the central frequency and bandwidth of the noise measurements reported?”

The noise measurement setup indeed includes a resonant LC tank. Detailed information are now provided in the new Supplementary Fig. S3 and Supplementary Note 2, including the precisions specifically requested by the reviewer (resonant frequency, used frequency window). The presence of a LC tank circuit is now also specifically mentioned at the end of the Methods section in the article.

“The clear discussion of accuracy and precision for CBO is not replicated in the discussion of quantum shot noise. Factors limiting accuracy are not properly discussed e.g. what is the accuracy with which the transmission coefficient of the QPC is determined? It would also be helpful to explicitly state acquisition times needed to achieve the reported precision. The section on dynamical coulomb blockade thermometry is more technical, and hard to follow in this account.”

Regarding the discussion on the possible artifacts for the quantum shot noise thermometry: Although the accuracy on the experimental determination of τ has no direct effect on the extracted temperature, a possible artifact that was previously not discussed is the presence of a voltage bias dependence $\tau(V_{dc})$. This effect is minimal at $\tau \sim 0.5$, as this correspond to a maximum of $\tau(1-\tau)$ in Eq. 1. In practice, $\tau(V_{dc})$ is simultaneously measured to ascertain a negligible voltage bias dependence. This possible artifact/difficulty is now pointed out in the Quantum shot noise thermometry paragraph of the section “Discussion”.

“The details on how the sample is thermalized to the mixing chamber should be simply and explicitly stated; it is difficult to extract this crucial information from the technical details provided.”

The key thermal anchoring is performed by dipping insulated copper wires into silver epoxy very close to the sample. We modified the second paragraph of the section “Discussion” to better put forward this information.

"I question the value of including the section of p5 on "electronic temperature vs. experimental condition". Fig 3. plots the electronic temperature vs power to the mixing chamber. To compare the temperature measured in the mesoscopic device with that measured by a reliable independent thermometer (not a secondary resistance thermometer) on the mixing chamber over a wide temperature range would be valuable, but this was not available. [Elsewhere the paper concentrates on a cross-calibration at a single temperature]. Here, merely to extract the heat leak to the mixing chamber as a function of magnetic field is just not of sufficient interest for the main text of a Nature Communication. [The caption to Fig 3 refers to "electronic" temperature. Measured by which technique?]"

As pointed out by the reviewer, the section "Electronic temperature vs experimental conditions" does not provide new information regarding the thermometry methods. However, in the context of a rapidly growing use of cryogen-free dilution refrigerators, it seemed to us of significant broad interest to show how the associated vibrations combined with a magnetic field do limit the base temperature in these systems. Regarding the thermometry method used for the displayed temperature in Fig 3, as now explicitly indicated (second sentence of the section) we used quantum shot noise thermometry up to 35 mK and then the readings of our standard RuO₂ thermometer. Indeed, we find at these "high" temperatures that the two methods give identical values (but the RuO₂ thermometer is much faster).

"Finally, to reinforce an earlier point, no detail is given on the noise thermometry set-up in this manuscript. I recommend adding this detail, in main text and in supplementary information. The criterion to provide sufficient information that the measurement could be reproduced, particularly important if the claim is to demonstrate new thermometry techniques, is not satisfied by this manuscript."

As detailed above, many changes have been made to address the main request of the reviewer to provide more technical details. In particular, we have added the new Supplementary Fig. S3 and Supplementary Note 2, and switched to show the experimental excess noise with respect to $V_{dc}=0$ (instead of using a confusing offset).

"It is also worth stating that the reliance on being in the quantum Hall regime probably makes this technique less generic and more niche than the impression the reader would get from reading the abstract. Perhaps the abstract should reflect this."

As now explicitly indicated in the article, being in the quantum Hall regime is not essential. See corresponding answer above.

“To repeat, I recommend publication subject to the above issues being satisfactorily addressed.”

Detailed list of changes:

Article:

- Addresses were updated (the LPN will form a new lab together with IEF, called C2N, starting June 2016).
- Figure 1a: the etched areas are now shown slightly brighter.
- Page 1, right column, first sentence: “obtained in a large array of ~ 100 μm wide metallic islands connected through tunnel junctions” was replaced by “obtained in a large array of 600 metallic islands, each ~ 100 μm wide and interconnected by tunnel junctions”.
- Page 1, last sentence: “quantum shot noise” was replaced by “current fluctuations”
- Page 2, second subtitle “Electronic quantum shot noise” was replaced by “Electronic current fluctuations”.
- Page 2, right column, first paragraph: We have added the sentence “Note that the quantum Hall effect is not necessary for the investigated primary thermometers (although it allows eliminating possible heating artifacts in the quantum shot noise thermometry, see Discussion).”
- Page 2, section “Electronic current fluctuations”: Several modifications were made as we now use the simpler excess noise ΔS_I , and also to point the new supplementary information material and to put forward that there is no need to separately calibrate the amplification chain to extract the electronic temperature.
- Figure 2a, top panel: “ S_I ” of the y-axis label was replaced by “ ΔS_I ”, and the data now correspond to $\Delta S_I = S_I(V) - S_I(V=0)$.
- Caption of Figure 2a: “measured spectral density of the current fluctuations” was replaced by “measured excess spectral density of the current fluctuations”, and “The continuous (dashed) line is the calculated quantum shot noise for $T_N=6.0$ mK ($T_N=0$)” was replaced by “The red continuous (dashed) line is the calculated excess current fluctuations for $T_N=6.0$ mK ($T_N=0$, with a matching negative offset).”
- Page 4, section “Coulomb blockade oscillations”, end of paragraph below Eq. 2: The two following sentences were added “Note that the Coulomb blockade thermometry is possible only with tunnel contacts. In the presence of connected conduction channels with large transmission probabilities, the quantum fluctuations of the island's charge average out Coulomb oscillations and thereby impede the Coulomb blockade thermometry (see Ref. 35 for a characterization of charge quantization versus transmission probability on the same device).”
- Page 4, section “Coulomb blockade oscillations”, last paragraph: “(corresponding to the repeated measurements of 14 adjacent peaks)” was replaced by “The 222 sweeps are distributed among 14 adjacent Coulomb peaks, spreading over 10 mV in gate voltage. We

find the same electronic temperature, at experimental accuracy, for the different Coulomb peaks and also for the 15 or 16 measurements of each peak”.

- Page 5, Section “Electronic temperature vs experimental conditions”: We have added the sentence “Note that T is here obtained from quantum shot noise thermometry up to 35 mK, and from the identical but faster readings of our standard RuO₂ thermometer at higher temperatures.”
- Page 6, second paragraph of “Discussion”: the sentence “The electrical measurement lines inside the inner stainless steel (SS) shield, closest to the sample, are insulated copper wires dipped into silver epoxy, in order to enhance their thermal anchoring to the dilution refrigerator.” was replaced by “The most important thermal anchoring of the measurement lines at base temperature is performed by dipping insulated copper wires into silver epoxy very close to the sample, inside the inner stainless steel (SS) shield.”
- Page 6, right column, discussion of quantum shot noise thermometry: We now mention that a separate calibration of the amplification setup is not necessary. We also discuss another possible artifact, the voltage bias dependence of the quantum conductor’s transmission probabilities.
- Methods, last sentence: “Regarding the noise measurement setup, see the detailed description in the supplementary material of Ref. 11” was replaced by “The noise measurement setup includes a home-made cryogenic preamplifier and a L-C tank circuit of resonant frequency 0.84~MHz, see the online Supplementary Information and also the supplementary material of Ref. 11 for a more detailed description.”
- New references [5,35].

Supplementary information:

- Supplementary Figure 1a: the etched areas are now shown slightly brighter.
- New Supplementary Figure S3.
- New Supplementary Note 2.

Reviewers' Comments:

Reviewer #1 (Remarks to the Author)

Dear Editor,

In the revised manuscript authors have addressed in detail all of my comments. I have read the comments of the other two referees and I thought the authors have addressed those comments as well.

I think this paper will become an important reference for electronic thermalization and temperature measurement in dilution refrigerators based on modern dry platforms and the papers should be published without any further delay.

Reviewer #3 (Remarks to the Author)

In the revised manuscript, the authors have made considerable efforts to address the comments of referees. It is interesting to note the overlap between the concerns of different reviewers. The clarity of the paper is improved and this paper should be published. Since I don't want to impede that process the following remarks should be taken as suggestive only.

I am still not entirely satisfied that the authors have adequately addressed the need to explain the article to readers who are not specialists. Since this is an article about thermometry, and is published in Nature Comms, bridging the expertise divide is important. The long (more than 20 years) history of shot noise measurements on mesoscopics and quantum hall devices will not be familiar to many readers.

Despite digging, quite extensively, into the literature devoted to the measurement of shot noise in semiconductor systems (from Heiblum, Glattli and Kobayashi groups), I am still not clear how the required noise floor is achieved in the work of Iftikhar et al., given the noise characteristics of the cryogenic amplifier. I do recognise that such a noise floor is quite common in this community.

My point is that the comment in my review "How the noise precision of $10^{-30} \text{ A}^2/\text{Hz}$ is obtained should be explicitly discussed" remains unaddressed. My suggestion is that the required information/noise calculation should be given in the caption to Fig S3.

[There is a clear mismatch between the noise precision of $10^{-31} \text{ A}^2/\text{Hz}$, and resistor to convert current to voltage noise of $4.3 \text{ k}\Omega$, and an amplifier of noise $0.2 \text{ nV}/\sqrt{\text{Hz}}$. If I was told the quality factor of the tuned circuit, that would help. I am missing something, and so I suspect will many other readers. Please explain.]

Reznikov et al. (Phys.Rev.Lett 75, 3340 (1995)) address such an issue quite well in their case, and clearly describe their scheme.

Just to note also that I also remain unconvinced that the quantum hall effect does not play a central role.

Surely the way that a quantum hall resistance converts a current to a voltage is intrinsically noiseless (given the nature of the resistivity tensor on a quantum hall plateau). Replacing it by a resistor would add Johnson-Nyquist noise.

I stand by my remark that the material on p5, basically telling us that the heat leak is magnetic field dependent, is not appropriate content for Nature Comms.

We would like to thank the reviewers for their time and remarks on our revised manuscript.

Our answers to the specific points raised by the reviewers are inserted within their comments below.

Reviewer #1:

“ Dear Editor, In the revised manuscript authors have addressed in detail all of my comments. I have read the comments of the other two referees and I thought the authors have addressed those comments as well. I think this paper will become an important reference for electronic thermalization and temperature measurement in dilution refrigerators based on modern dry platforms and the papers should be published without any further delay. “

We thank the reviewer for his/her positive appreciation of our work and for his/her prediction of its future impact.

Reviewer #3:

“In the revised manuscript, the authors have made considerable efforts to address the comments of referees. It is interesting to note the overlap between the concerns of different reviewers. The clarity of the paper is improved and this paper should be published. Since I don't want to impede that process the following remarks should be taken as suggestive only.”

We thank the reviewer for acknowledging our “considerable efforts” in this revised manuscript and we definitely recognize the improvements the reviewers ‘comments lead to.

“I am still not entirely satisfied that the authors have adequately addressed the need to explain the article to readers who are not specialists. Since this is an article about thermometry, and is published in Nature Comms, bridging the expertise divide is important. The long (more than 20 years) history of shot noise measurements on mesoscopics and quantum hall devices will not be familiar to many readers. Despite digging, quite extensively, into the literature devoted to the measurement of shot noise in semiconductor systems (from Heiblum, Glattli and Kobayashi groups), I am still not clear how the required noise floor is achieved in the work of Iftikhar et al., given the noise characteristics of the cryogenic amplifier. I do recognise that such a noise floor is quite common in this community. My point is that the comment in my review "How the noise precision of $10^{(-30)} \text{ A}^2/\text{Hz}$ is obtained should be explicitly discussed" remains unaddressed. My suggestion is that the required information/noise calculation should be given in the caption to Fig S3. [There is a clear mismatch between the noise precision of $10^{(-31)} \text{ A}^2/\text{Hz}$, and resistor to convert current to voltage noise of 4.3kohm, and an amplifier of noise 0.2nV/rootHz. If I was told the quality factor of the tuned circuit, that would help. I am missing something, and so I suspect will many other readers. Please explain.] Reznikov et al. (Phys.Rev.Lett 75, 3340 (1995)) address such an issue quite well in their case, and clearly describe their scheme.”

The experimental relative precision on the noise spectral density $\delta S/S$ decreases as $1/\sqrt{N}$ with N the number of measurements. When measuring a white noise, N is proportional to the product of the time of integration and the measured frequency window; increasing one of this parameter then results in an increase of N and a better (smaller) precision. We now provide the values of these parameters as well as their impact on the noise resolution in the caption of Fig S3.

“ Just to note also that I also remain unconvinced that the quantum hall effect does not play a central role. Surely the way that a quantum hall resistance converts a current to a voltage is intrinsically noiseless (given the nature of the resistivity tensor on a quantum hall plateau). Replacing it by a resistor would add Johnson-Nyquist noise. I stand by my remark that the material on p5, basically telling us that the heat leak is magnetic field dependent, is not appropriate content for Nature Comms. “

We note that we cannot see any crucial difference for the current to voltage conversion between using a quantum Hall resistance and using a standard (macroscopic, eg CMS) resistance, except the remarkable precision of the quantum Hall resistance. In particular, quantum Hall resistances are not noiseless since the standard Johnson-Nyquist $4kT/R$ noise also applies to them (with $R=h/\nu e^2$ and ν the filling factor).

Detailed list of changes:

Article:

- Addresses were modified to use “université Paris Diderot” instead of “univ Paris Diderot”
- Page 1, first column, first sentence: “Novel” was removed.
- Last page : a section “Competing Financial Interests” was added after the section “Author contributions”
- Figure 1: the scale bar label was removed.
- Caption of Figure 1, **a**: the length of the scale bar was added.
- Caption of Figure 1, **a**: “by field effect” was added.
- Caption of Figure 1, **b**: “Using the switches,” was added while “by field effect” was removed.
- The reference number 35 was updated with the DOI of the accepted publication in Nature: “in Press, doi: 10.1038/nature19072”

Supplementary information:

- Title, author list and affiliations removed from the title.
- Caption of Figure S3 rewritten to give further details in response to reviewer 3.